# The Effect of Dry Hopping Efficiency on β-Myrcene Dissolution into Beer

**DOI:** 10.3390/plants11081043

**Published:** 2022-04-12

**Authors:** Rozália Veronika Salamon, Adriana Dabija, Ágota Ferencz, György Tankó, Marius Eduard Ciocan, Georgiana Gabriela Codină

**Affiliations:** 1Department of Food Science, Faculty of Economics, Socio-Human Sciences and Engineering, Sapientia Hungarian University of Transylvania, Piata Libertatii no. 1, 530104 Miercurea Ciuc, Romania; salamonrozalia@uni.sapientia.ro (R.V.S.); ferenczagota@uni.sapientia.ro (Á.F.); 2Faculty of Food Engineering, Stefan cel Mare University of Suceava, 720229 Suceava, Romania; adriana.dabija@fia.usv.ro (A.D.); marius_ec@yahoo.com (M.E.C.); 3Doctoral School of Food Science, Magyar Agár-és Élettudományi Egyetem, Str. Villányi 29-43, 1118 Budapest, Hungary; gyorgy_tanko@yahoo.com

**Keywords:** fermentation, beer, hops, extraction, gas chromatography

## Abstract

The production of heavily hopped beers, such as Indian Pale Ale (IPA) styles, has been gaining momentum in recent years in the Central European markets. To this end, the dry hopping process is becoming increasingly popular, mostly in microbreweries, but also with larger manufacturers. In our research, we investigated the dissolution rate of the main volatile component of hops, β-myrcene with a modified dry hopping method. Following the primary fermentation, we applied the dry hopping process, where the weighed hops were chopped and blended into a container with 0.5 L of beer and later added to the young beer. During the dry hopping process, we determined various important parameters of the beer, and we repeated the same measurements for the bottled beer. In the first 96 h of the dry hopping process, we monitored the concentration of β-myrcene so that we managed to determine the dissolution rate constant (k = 0.1946 h^−1^). The β-myrcene concentration stabilizes after 44 h in the fermenter. At the same time, measurements were conducted for bitterness, pH, CO_2_ and alcohol content, extract and density during the process. Our experiment demonstrates that a new method of dry hopping provides a much higher concentration of β-myrcene (215 μg/L) than other methods indicated in former studies in the field. A health and safety assessment of β-myrcene was also made and we determined what the safe amount of β-myrcene ingested with IPA beer is. Our modified process was successful, we were able to determine the dissolution rate of β-myrcene, and the recommended daily intake of IPA beer with particular reference to β-myrcene.

## 1. Introduction

Beer production is an age-old business, with the most consumed beer styles being lightly hopped varieties. To this end, hops have been traditionally used to influence the microbial and flavor stability of beer for centuries, where the main function of hops has been to add flavor, bitterness and—to a lesser extent—some aroma to the beer. Up until the mid-20th century, hops were used primarily for their bitterness [1]. The most common hop usage technique to this day is to add hops in the boiling kettle, or hopping kettle, during the brewing process. Hops added at the beginning of boiling are expected to impart most of their bittering potential, the utilization for whole cone hops and pellets ranges from 24–40% (in IBU) in a 60 min addition (depending on the kettle design, temperature and wort strength) [2,3]. Ever since the 1970s in the USA, and over the past 5–10 years in Central Europe, brewers have been producing beer styles with more intense aromas. Usually, microbreweries tend to utilize a technique called “aroma” hop additions, where hops are added to enhance or preserve the aromatic properties of hops without trying to impart hop bitterness. Less bitterness utilization occurs (5–20%) when hops are added at the end of boiling (late kettle hopping) and/or later in the brewing process, such as in the whirlpool (whirlpool hopping) or in the hop-back (hop-back hopping) [2,3]. It is not unheard of to cool the wort (75–95 °C) either before the whirlpool or in an auxiliary vessel (hop-back) to further improve hop aroma retention and minimize hop acid isomerization. The dry hopping procedure is the most popular technique used by breweries to prevent isohumulone-derived hop bitterness.

The first reported use of dry hopping took place in the British Empire and its overseas colonies, where brewers used dry hopping to increase both the microbial and flavor stability of beer [4]. Nowadays the average dry-hopping rate in our day has increased dramatically: most brewers use two to three times greater amounts of hops than the reported historical rates [3,5]. Broadly, dry hopping is defined as a cold extraction process (4–20 °C) of non-volatile and volatile chemicals from hops into an alcoholic solution without increasing product bitterness [1]. There are several dry hopping procedures used by breweries [6], the most notable being the traditional dry hopping method, where hop pellets or whole hop cones are added into the fermentation vessel, either in bags or without a bag. This procedure can also be performed by adding the hops into one vessel and pumping the beer onto it. This method has many drawbacks, including oxidation and infections. Newer methods include the use of “hopguns”, ancillary vessels that use CO_2_ overpressure or pumps to move the beer through the hops, maximize the extraction efficiency and minimize the risks of traditional methods [6]. Drawbacks include the high acquisition costs and the danger of hops clogging up the stubs of the vessels. Several studies have shown that the dry hopping procedure increases beer bitterness slightly. Humulinones (oxidized α-acids) have been identified as important compounds that increase beer bitterness [1,7,8,9,10]. The total essential oil fraction of hops ranges from 0.5 to 4.0% w/w depending on the variety; these compounds are regarded as the main source of hop aroma. Recently, Rettberg et al. [11] published a comprehensive review on the current understanding of the drivers of hop aroma and their subsequent analysis in hops and beer. The total essential oil content of the hops is the main analytical indicator used to determine the aroma intensity and quality of hops being used in the dry hopping procedure. However, Vollmer and Shellhammer [12] observed that total oil content was not an effective indicator of hop aroma potential in the process of dry hopping and suggested that the composition of hop essential oil was more important. The composition of hop essential oil is estimated to be made up of hundreds of compounds [1]. The compounds that make up the hop essential oils can be characterized into three general groups: hydrocarbons (mainly monoterpenes and sesquiterpenes), oxygenated compounds and sulfur-containing compounds [1,11]. Monoterpenes and sesquiterpenes make up 70–80% of hop essential oil, depending on the variety, and a number of studies have demonstrated that they are important in creating the aroma of raw hop [13,14,15]. Due to their physical–chemical properties, monoterpenes and sesquiterpenes are not typically found in beer at concentrations above their detection thresholds (β-myrcene, 350 µg/L; α-humulene, 450 µg/L; and β-caryophyllene, 230 µg/L) [1] and, thus, are unlikely to contribute to dry-hop aroma. β-Myrcene (7-methyl-3-methylideneocta-1,6-diene), is often referred to as “fresh hop oil”, its taste and aroma are extremely diverse, herbaceous, resinous, green, balmy and slightly metallic. This is the main component of many hop varieties. Low boiling point and strong volatility are characteristic of β-myrcene, therefore, it evaporates during boiling, it becomes unstable at high temperatures or it oxidizes very quickly, thus it is best to use in dry hopping [16].

β-Myrcene has shown promising health benefits in many animal studies. However, studies conducted on humans are lacking. The main reported biological properties of β-myrcene—anxiolytic, antioxidant, anti-ageing, anti-inflammatory and analgesic properties—are discussed [17]. The National Toxicology Programs, USA (NTP) report links β-myrcene to neoplasms of the kidney in male rats and liver cancer in male mice [18]. The daily per capita intake (eaters only) of β-myrcene was estimated as being 164 µg corresponding to 3 µg/kg body weight (bw) [19]. The wide application of β-myrcene in industry and for domestic use is coupled with safety concerns. Overall, evidence has shown that the estimated daily safe intake of β-myrcene is 1.23 μg/kg bw/day for a 60 kg person [18]. Only a few researchers have quantified the β-myrcene content of beers, which can vary greatly depending on the recipe used (80–470 µg/L), but it can be said that IPA beers have the highest myrcene content due to dry hopping [20,21,22,23]. It is therefore very important to know the recommended daily intake of IPA beer that is considered safe in terms of myrcene content.

Quantitative and qualitative determination of β-myrcene can be carried out using various analytical techniques such as GC-FID, GC-MS [24] and HS-GC-MS [25].

The aim of this study was to determine the amount of β-myrcene volatile compound, the main component of the hop oil, during dry hopping and to calculate daily intake of IPA beer that is still safe amount regarding its β-myrcene content. In addition, the physicochemical changes that occur in beer samples during dry hopping were also analyzed and discussed.

## 2. Results

### 2.1. Chemical Properties of the Final Product

According to the Beer Judge Certification Program (BJCP) maintained by the BJCP Style Guidelines Committee [26] the finished beer fits into the 21B special black IPA category, having a definitive hoppy and bitter taste and aroma. The measured original extract, final extract, alcohol content, CO_2_ and O_2_ content, turbidity, color and bitterness are presented in Table 1.

### 2.2. pH, Bitterness and Density Change during Dry Hopping

Change in the bitterness and density values can be seen in Figure 1 and Figure 2. From the start of the dry hopping procedure, a slow but steady increase in bitterness took place, starting from 54.2 IBU to 64.3 IBU. A significant (*p* < 0.05) bitterness increase of 15.6% was obtained during dry hopping. This increase was much higher than that obtained by others. Cocuzza and Mitter [27] report only a 6% increase in bitterness after dry hopping. The pH value increase was insignificant (*p* < 0.05) from 4.59 to 4.61 (0.43%), as shown in Figure 3. These data do not correlate with data collected by other researchers as they report a significant increase in pH value after dry hopping [27,28]. CO_2_ content was also measured, with 3.56 g/L in the young beer and 4.48 g/L in the bottled product.

### 2.3. Alcohol Content and Extract Variations during Dry Hopping

Mitter and Cocuzza also report that there is a small amount of alcohol concentration increase during the dry hopping procedure (7%) [6]. We also measured an insignificant increase (*p* < 0.05) in alcohol concentration (Figure 4) during dry hopping at 22 h mark, reaching a maximum concentration of 6.12 *v*/*v*%, followed by a decrease to 5.8 *v*/*v*% at 92 h. In the course of the dry hopping process the trend for density correlates with the alcohol content changes in the beer. We have measured a drop in density from 1.0255 g/cm^3^ to 1.0235 g/cm^3^ (Figure 2), similar data being also reported by Hauser et al. [29].

### 2.4. β-Myrcene Concentration

Figure 5 contains data from the first 84 h of dry hopping. β-Myrcene concentration reaches a maximum of 301 μg/L at 34 h, followed by a decrease in concentration and levelling out at 215 μg/L starting from 42 h.

The dissolution efficiency was also calculated, based on the equation below. The concentration of β-myrcene in the young beer (0 μg/L) is subtracted from the concentration of the measured β-myrcene concentration and it is divided by the maximum theoretical concentration of β-myrcene found in hops (3760 μg/L) calculated based on used hops material datasheet obtained from the producer (Yakimachief hops).
$$\mathrm{Dissolution}\;\mathrm{efficiency}=\frac{\mathrm{measured}\;\beta -\mathrm{myrcene}\;\mathrm{conc}.\;\mathrm{in}\;\mathrm{dry}\;\mathrm{hopped}\;\mathrm{beer}\;-\;\beta -\mathrm{myrcene}\;\mathrm{conc}.\;\mathrm{in}\;\mathrm{young}\;\mathrm{beer}\;}{\max.\;\mathrm{theoretical}\;\mathrm{conc}.\;\mathrm{of}\;\beta -\mathrm{myrcene}\;\mathrm{added}\;\mathrm{with}\;\mathrm{hops}\;\mathrm{during}\;\mathrm{dry}\;\mathrm{hopping}}$$

The dissolution efficiency change can be found in Figure 6. Maximum dissolution is reached at 34 h, with a corresponding value of 0.08. This value is lower by 20% than other data found in the literature by Forster and Gahr, [30], who reported a dissolution efficiency of 0.1.

An exponential function was established as a model to calculate the dissolution speed constant, based on the maximum concentration measured at 34 h.

By curve fitting on the experimental results the model parameters were determined (Figure 7): maximum concentrations (c_max_ = 301 μg/L), dissolution rate constant (k = 0.1946 h^−1^), with correlation coefficient R^2^ = 0.873. For the dissolution rate the following model was adopted c = c_max_(1 − e^−kτ^). The measured and calculated β-myrcene concentrations were plotted in Figure 8 (R^2^ = 0.9799). The differences were probably given by the inhomogeneity of the solution due to the size of the fermentation vessel.

β-Myrcene concentration does not drop below 215 μg/L even after bottling, remaining close to the levels measured at the end of the dry hopping procedure, at 96 h and 228 μg/L. This concentration fits the data that can be found in the literature (2–470 μg/L) [20,21,22,23,26].

Considering the β-myrcene concentration of the produced IPA beer a 60 kg person can consume 0.35 L per day without exceeding the recommended safe dose of β-myrcene.

## 3. Discussions

Dry hopping is one of the last trends in beer production. It is defined as a cold extraction of hops components in an alcoholic solution [4,5]. As a consequence, the beer becomes richer in hop flavor and more stable from the point of view of microbiology. According to Shopska et al. [31] the extraction of the hop components depends on the beer’s physical–chemical parameters which were determined in this study. According to our data, during dry hopping, the physical-chemical parameters such as original extract, final extract, original density, alcohol, carbon dioxide, oxygen, turbidity, pH and bitterness values increased whereas the final extract, final density and color decreased. The variation of these factors was due to the biochemical processes taking place during this stage of fermentation [31,32] with certain impact on the quantitative and qualitative composition of the final beer. However, because the fermentation process is slow, after dry hopping, the variations of the parameter’s original extract, final extract, original density and color for beer samples before and after hopping were insignificant (*p* < 0.05). After the yeast is pitched and dry hop is introduced, the fermentation process continues based on a small amount of yeast cells remaining in the fermentation vessel, which allow the fermentation process of the existing sugars in the young beer to continue. According to our data, this will lead to a decrease in the final extract up to 3.76% during dry hopping. The remaining yeasts use the amino acids in young beer as a source of nitrogen and use glucose, maltose and maltotriose as a source of carbon and energy, with the generation of ethyl alcohol, carbon dioxide and other by-products of fermentation [33,34,35]. Therefore, the alcohol content in dry hopped beer increased by 3.5% and the carbon dioxide content increased by 25.84% during the procedure. Speers [36] concludes that the amount of carbon dioxide content in beer depends on pressure, temperature and beer composition. According to our data, the significant increase (*p* < 0.05) in carbon dioxide content is also due to the fact that the fermentation process takes place in a closed vessel, at a pressure that allows it to accumulate in beer. Moreover, in our experiment, the fermentation temperature decreased from 18 °C to 8 °C, which favors the significant increase in CO_2_ content. The physical absorption capacity of carbon dioxide is also influenced by the composition of the beer, especially by the distribution of colloids that have a very large contact surface. The surface active of the colloids molecules enter into the gas wall and form a framework that holds it together [37]. By using dry hopping, the amount of colloids in beer increases due to the high content of colloids from hops leading to an increase in the absorption of carbon dioxide.

After dry hopping the beer’s turbidity increased significantly (*p* < 0.05) up to 45.84% compared to the young beer. Turbidity in beer is produced by protein combinations, polyphenols, hop resins, yeast cells and sometimes other microorganisms [38]. According to Kahle et al. [39] the rate and intensity of turbidity formation are proportional to the degree of polymerization of the polyphenols. Between them and proteins, complex combinations are formed by bonds between the oxygen of the peptide groups and the hydroxyl groups of the polyphenols, as well as by covalent bonds. The addition of dry hopping into the young beer led to an increase in the content of hop resins and tannins. The tannin substances in hops are a complex mixture of chemical compounds ranging from small molecule polyphenols to macromolecular polyphenols with varying degrees of polymerization and condensation [40].

The bitterness value of beer increased significantly (*p* < 0.05) with almost 18% during the dry hopping. This may be due to the fact that the hop vegetal material can adsorb isomerized alpha-acids, allowing their oxidized form (water-soluble humulinones) to make a significant contribution to bitterness. Humulinones are insoluble at high temperatures, therefore their impact on beer bitterness in traditional hop additions is negligible [41]. According to Algazzali et al. [8] it seems that humulinones are almost 66% as bitter as isoalpha-acids contributing significantly to the beer bitterness profile. Ferreira et al. [10] concluded that after dry hopping, humulinones were responsible for up to 28% of beer bitterness, being a good marker for the quality of beer. Their effect is less significant for the beer obtained through the traditional hoping process due to the fact that these components are highly degraded during boiling and fermentation when they are absorbed into yeast. Furthermore, it has also been reported that in dry hopping polyphenol extraction from hop content and oxidation products of lupulones (hulupones) can also contribute to the bitterness and astringency of beer [42,43,44]. Even if some polyphenols and humulinones have been reported as imparting bitterness, it seems that their contribution has been perceived as different. Parkin and Shellhammer [44] report that the perceived bitterness was highly correlated with humulinone and less correlated with polyphenols content.

The increased oxygen content in the finished sample is due to the introduction of oxygen during dry hopping. This is one of the major problems encountered when using this hop technique because oxygen may be one of the enemies of retaining the bright, fresh and vibrant hop character [45]. Fortunately, in our case the oxygen increase was not significant (*p* < 0.05).

During dry hopping, the pH value does not vary significantly (*p* < 0.05). However, this is an essential parameter in the technology of obtaining beer. Our data are in disagreement with the data reported by others, where a significant increase in pH value after dry hopping was observed [23,28]. However, due to the amino acid’s consumption by yeasts during fermentation the pH value of beer tended to decrease. This linearity of pH value after dry hopping may be due to the hop’s leaf material being extremely water-soluble, which tends to increase the pH value, but the exact cause of this increase has not been determined yet [9,46,47].

β-Myrcene was not detected in the young beer sample whereas after dry hopping it value increased in the first 34 h. This is one of the components of the lupine essential oils that give hop cones their characteristic scent, namely the aroma of beer. Essential oils represent 0.3–1.5% of the dry matter of hops. The aroma of fresh hops is mainly determined by β-myrcene, a monoterpene [48,49,50]. The essential oils in hops are not very soluble in water, they can be transformed during fermentation. Therefore, the amount of β-myrcene was 0 μg/L in the young beer. With dry hopping the essential oils were absorbed in the beer. We observed an increase in the initial concentration of β-myrcene in the beer followed by a decrease in its value at later points of the measurements. As β-myrcene is a small hydrocarbon molecule, its structure resembles the acyl chain (terminal part) of the membrane steroids. Therefore, when β-myrcene migrates from the beer solution it can pack in the voids between the palmitoyl chains and change their orientation at the yeast/beer interface [48,51,52,53]. The non-polar β-myrcene may attach to the non-polar surface of the yeast cells [54,55,56]. This leads to a decrease in the concentration of β-myrcene during the fermentation of young beer. The yeast cells sediment during fermentation also contributes to this decrease. The pH of beer may also contribute to a lower level of solubilization of β-myrcene. A slight increase in its value led to a decrease in the solubilization of essential oils from hops and implicitly of the β-myrcene. According to our data, the pH value after dry hopping was constant, thus it does not influence solubility of β-myrcene in beer.

The success of dry hopping depends on the method [6,7], volume of the used fermentation tank and the geometry. The different heights and proportions of the containers may cause the discrepancy, as they have a different effect on the laws of physics [57].

Maximum dissolution and dissolution speed constant are very important parameters when considering the development of IPA beer production to an industrial level, as they help the brewmaster decide how long the dry hopping process should be and how much hops should be used to achieve excellent organoleptic properties in the final product. It is also important to know how long it takes for the volatile compound, such as β-myrcene, to reach concentration equilibrium in a given volume. From this point of view, our results are important, as no similar data have been found in other research.

We found data on the recommended daily intake of β-myrcene [17,18,19], but not on the quantity of beer that can be consumed safely. Since diet is considered to be the greatest source of human exposure to β-myrcene there is a paucity of information on the subject. As IPA beers may contain high concentrations of β-myrcene [20,21], we believe that our results (0,33 L/day) are completely new in this respect.

## 4. Materials and Methods

### 4.1. Raw Materials and Equipment Used

Based on our previous experience we chose a highly hopped beer style recipe, a black IPA, to conduct our experiments on. We produced the beer in lab-scale conditions with a minimum 50 L/brew semi-automatic equipment and fermented in 100 L steel fermenters, produced by ZipTech (Miskolc, Hungary). The wort was prepared from Pale ale, Münich, Carafa Special type 3, Dark Wheat and Chocolate Wheat malts obtained from Weyermann^®^ Specialty Malts company (Bamberg, Germany), Hallertauer Magnum, Simcoe, Cascade and Amarillo hops obtained from Yachima Chief (Yakima, WA, USA), 12 °dH pre-boiled tap water (Romania) and New England yeast (Lallemand Inc., Montreal, QC, Canada).

### 4.2. Wort Preparation

For mashing, we utilized 12 kg Pale Ale malt, 3.6 kg Münich malt, 0.72 kg Chocolate Wheat malt, 0.36 kg Carafa Speciale type 3 malt, 0.6 kg Dark Wheat malt and 0.36 Carafa malt with 54 L mashing wather. The pH of the mash was corrected to 5.2 with 5 mL of 1% ortho-phosphoric acid solution. Traditional infusion mashing was used with resting temperatures and times as follows: 65 °C for 60 min, 72 °C for 20 min and 78 °C for 2 min, followed by lautering and sparging with 45 L of 78 °C water. Wort boiling lasted for 60 min, 86 g of Hallertauer Magnum hops were added at the start of the boiling, 48 g of Simcoe, 48 g of Amarillo and 24 g of Cascade hops were added at 30 min after start of boiling, 24 g of Simcoe, 24 g of Amarillo and 12 g of Cascade hops were added at 60 min after boiling. Ten grams of Irish moss were added 10 min before boiling stop. The 60 L final wort was cooled to the inoculation temperature (18 °C) with a plate-type heat exchanger and transferred to the fermentation vessel. Seven grams of yeast (rehydrated in 50 mL 25 °C water) were pitched. The extract, pH, temperature and pressure of the fermenting beer were measured.

### 4.3. Dry Hopping

After the main fermentation was over the temperature of the vessel was lowered to 8 °C and the yeast was removed. We weighted the hops for the dry-hopping procedure, 5 g of Simcoe, 5 g of Amarillo and 5 g of Cascade were used. The fermentation vessel was depressurized by the pressure regulator, 0.5 L of beer was taken from the vessel and a mixer was used to mince (30 s) the hop pellets in the liquid. The mixture was then poured back onto the beer from the top opening of the fermentation vessel. The vessel was repressurized by using CO_2_ from the inlet at the bottom of the vessel and this method was used to blend the beer with the dry hops. One hundred milliliters of sample were taken every 2 h in the first 48 h of the experiment, followed by every 4 h for the next 48 h. All samples were taken in triplicate. After every sample the contents of the vessel were mixed by introducing CO_2_ at 9 L/min flow rate for 5 min. The samples were kept cooled at −20°C until the analytical procedures could be performed. Figure 9 shows the dry hopping procedure.

### 4.4. Analytical Measurements

#### 4.4.1. pH Measurements

The pH of the different ingredients and intermediate products were measured by using a WTW Inolab 720 table pH meter (Xylem Analytics, Ingolstadt, Germany), according to EBC 9.35/2004 standard [58]. Triplicate measurements were performed for all samples.

#### 4.4.2. Extract, Alcohol Content and Density Measurements

Extract, alcohol content, density, CO_2_ and O_2_ concentration and turbidity of the beer during the fermentation and dry hopping process was measured using an Anton Paar Alex 500 type analyzer (Anton Paar GmbH’s, Graz, Austria) and Anton Paar DMA 35 portable density meter (Anton Paar GmbH’s, Graz, Austria) according to EBC 9.43.2/2004 standard [59]. Triplicate measurements were performed for all samples.

#### 4.4.3. Color Measurement

The color of the beer was determined by a Hach Lange DM 6000 spectrophotometer (Hach Lange GmbH’s, Düsseldorf, Germany) using the EBC 9.6/2004 standard [60] at 275 nm wavelength. Triplicate measurements were performed for all samples.

#### 4.4.4. Bitterness Measurement

Beer bitterness was measured according to EBC 9.8/2020 standard [61], by centrifuging with Hettich Universal 320 R (Hettich GmbH & Co., Tuttlingen, Germany) the beer at 4000 rpm for 20 min at 20 °C, 1 mL of 3 M HCl and 20 mL isooctane were added to 10 mL beer sample and it was mixed with IKA KS 260 basic shaker (IKA^®^-Werke GmbH & Co., Staufen im Breisgau, Germany) at 450 rpm for 25 min. The samples were left in complete darkness for 30 min and were measured spectrophotometrically (Hach Lange DM 6000, Hach Lange GmbH’s, Düsseldorf, Germany) at λ = 275 nm wavelength. Triplicate measurements were performed for all samples.

#### 4.4.5. β-Myrcene Concentration Measurement

β-Myrcene content was analyzed using Headspace GC-FID equipment (Perkin-Elmer Clarus 580, PerkinElmer Inc., Waltham, MA, USA). The separation column was an Elite-wax (ETR) l = 30 m length, d = 0.32 mm diameter, df = 0.5 μm film thickness. The headspace temperature was set to 60 °C, the needle temperature to 80 °C and the transfer path to 100 °C. Injection time was 0.1 min, time under pressure 1 min, temperature regulation 20 min, separation time 25 min and the pressure of the nitrogen carrier gas was 35 psi (241 kPa). The detection parameters at the FID were 150 °C temperature set for the detector, the column start temperature was set to 55 °C, with an increase of 15 °C/min until 150 °C and stabilizing for 3 min. The carrier gas used was hydrogen (40 mL/min).

The concentration of β-myrcene standard solutions was: 250, 500, 750 and 1000 μg/L. The concentration of the stock solution was 268 mg/L. The stock solution was prepared as follows: 30 mL of beer from the fermentation vessel (not containing any β-myrcene and CO_2_) was poured into a 50 mL measuring flask, 18.8 μL of β-myrcene standard (Sigma-Aldrich, Darmstadt, Germany) was added to it and then the vessel was filled to level with beer.

During calibration an internal standard (IS) was also used, consisting of a 38.22 mg/L butanol in 5 *v*/*v*% ethanol solution. 99.9% purity ethanol was obtained from Merck, the butanol was obtained from Sigma Aldrich (HPLC grade).

To obtain the standard solutions 4 × 100 mL of beer was measured and β-myrcene stock solution was added to them (93.3, 186.6, 279.9 and 373.2 μL) with an automatic pipette, with an additional 800 μL of IS. The solution was homogenized, 5 mL was measured out into the headspace glass vials and they were measured in the gas chromatography equipment. The obtained calibration curve of β-myrcene is shown in Figure 10 and the typical chromatogram in Figure 11.

The method was validated using β-myrcene standard normalized by butanol internal standard. Specificity and selectivity were studied for the examination of the presence of interfering endogenous components. The result indicates that the retention time of β myrcene was about 2.41 and none of the impurities interfered in its assay. Linearity was studied by preparing standard solutions at different concentration levels. The linearity range was found to be 0.25 to 10 µg/mL, R^2^ = 0.9913.

Recovery was determined by beer samples spiked with a standard solution at three concentration levels and the % recovery was found in the range of 94.72 to 103.56%.

Intra- and interday variations were calculated to determine the precision of the method. The intraday variation was determined for six concentration levels covering the analyte calibration range. The interday variation was determined through the analysis of these standard solutions on three consecutive days. The range for the accuracy for both intraday and interday precision determinations was from 89.72 to 116.31% while the coefficient of variance in both was less than 7.5%.

All samples taken during the dry hopping procedure were in triplicate. Averages were calculated from the measurement results (with maximum standard deviation 7.73) after which a rolling average was also calculated, as seen in the Figure 5.

### 4.5. Statistical Analysis

All data were expressed as the means of the measurements ± standard deviation and were made in triplicates. The one-way analysis of variance (ANOVA) with Tukey’s test was employed to evaluate the differences considered significant at *p* < 0.05. Statistical tests and principal component analysis were made using XLSTAT software for Excel 2021 version (Addinsoft, New York, NY, USA).

## 5. Conclusions

The beer analyzed during our experiment fits well into the BCJP category based on its aspect, color, character, taste and aroma. The dry hopping procedure was successful, the most critical component (β-myrcene) concentration was measured at higher levels than those reported in the literature. We can conclude that by pre-blending the hops without using hop bags we achieved a slightly higher (8%) β-myrcene concentration in the beer. We observed an insignificant increase in pH and alcohol content, and a small decrease in extract and density, but an important increase in CO_2_ content and bitterness values.

We found that during dry hopping in a 100 L fermentation vessel, the maximum concentration of β-myrcene was reached after 34 h and the concentration equilibrium occurred after 42 h.

We also determined the dissolution rate constant of β-myrcene (k = 0.1946 h^−1^), which is an important data for industrial scale applications.

We can state that the β-myrcene concentration in bottled beer is 215 μg/L, which is close to the value found at the end of the dry hopping procedure (228 μg/L). Based on these results, we can conclude that responsible IPA beer consumption (0.33 L/day) does not pose a risk to human health in terms of β-myrcene.

## Figures and Tables

**Figure 1 plants-11-01043-f001:**
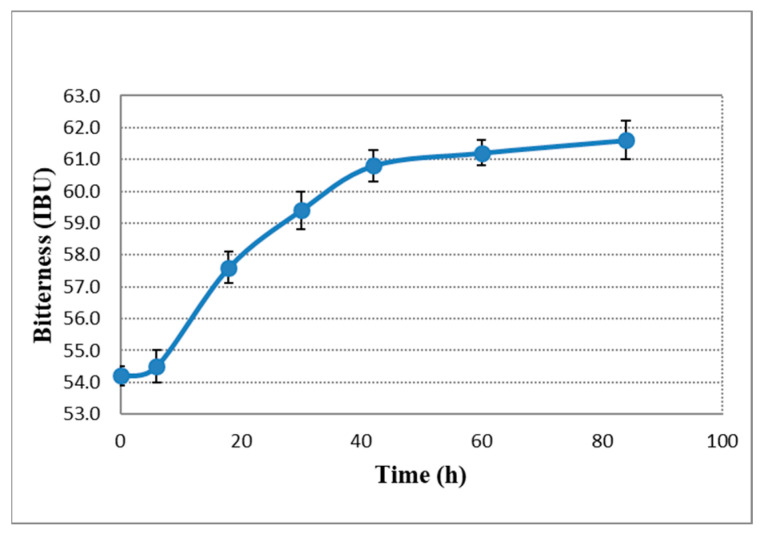
Bitterness increase in beer during dry hopping.

**Figure 2 plants-11-01043-f002:**
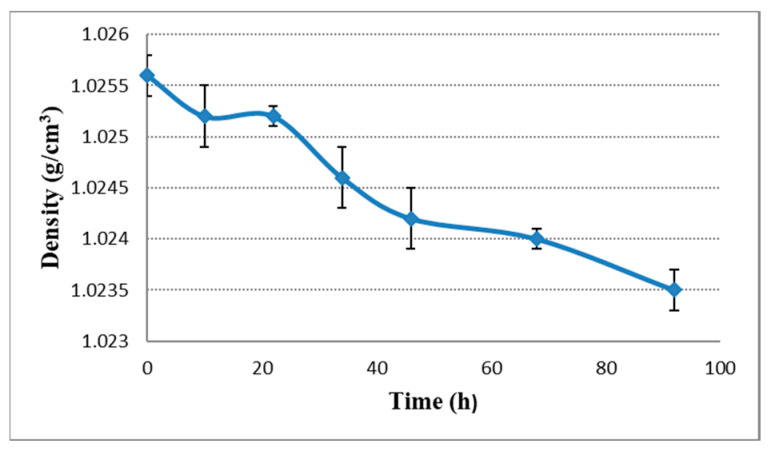
Density change during dry hopping in beer.

**Figure 3 plants-11-01043-f003:**
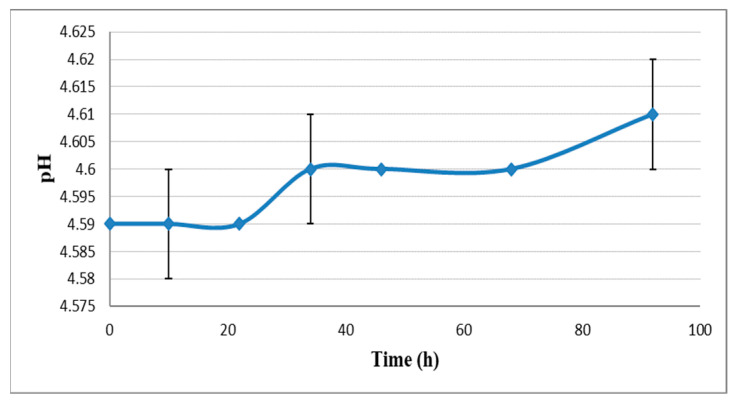
pH change during dry hopping in beer.

**Figure 4 plants-11-01043-f004:**
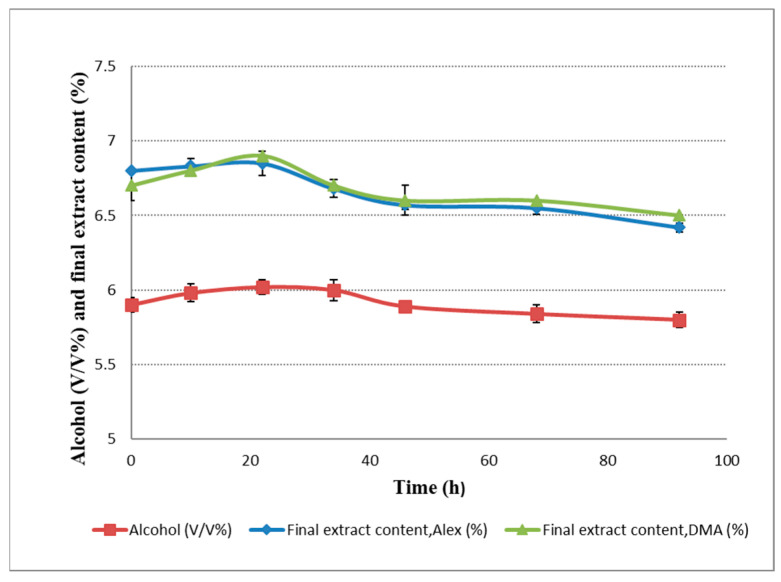
Alcohol and extract change during dry hopping in beer measured with Anton Paar Alex 500 type analyzer (Anton Paar GmbH’s, St Albans, UK), and DMA-Anton Paar DMA 35 portable density meter (Anton Paar GmbH’s, UK).

**Figure 5 plants-11-01043-f005:**
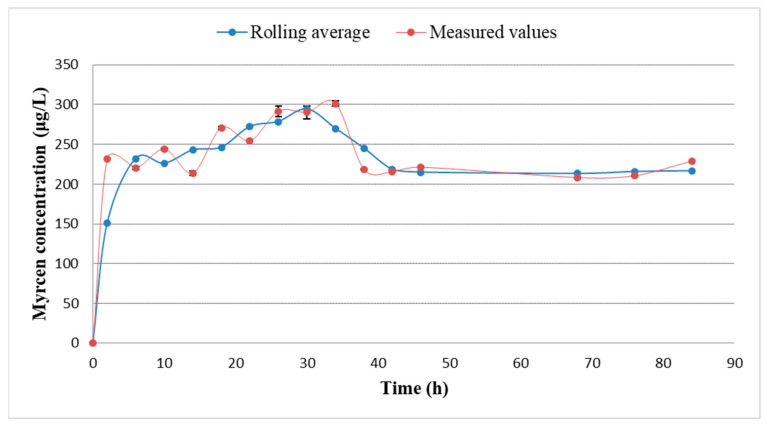
β-myrcene concentration change during dry hopping in beer.

**Figure 6 plants-11-01043-f006:**
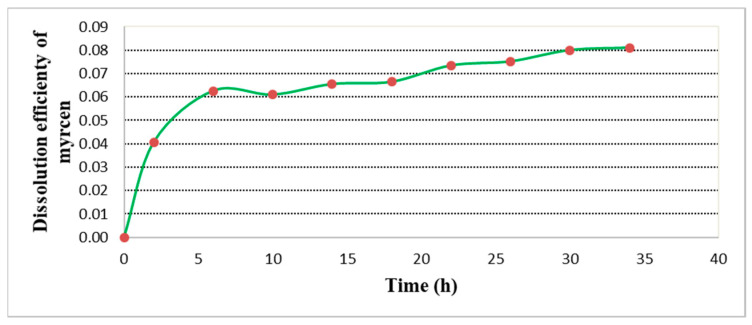
β-myrcene dissolution efficiency.

**Figure 7 plants-11-01043-f007:**
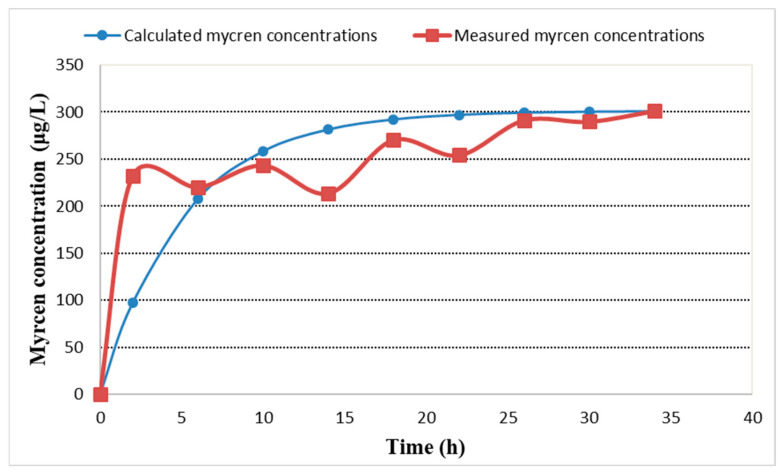
Recalculated β-myrcene concentrations based on the model.

**Figure 8 plants-11-01043-f008:**
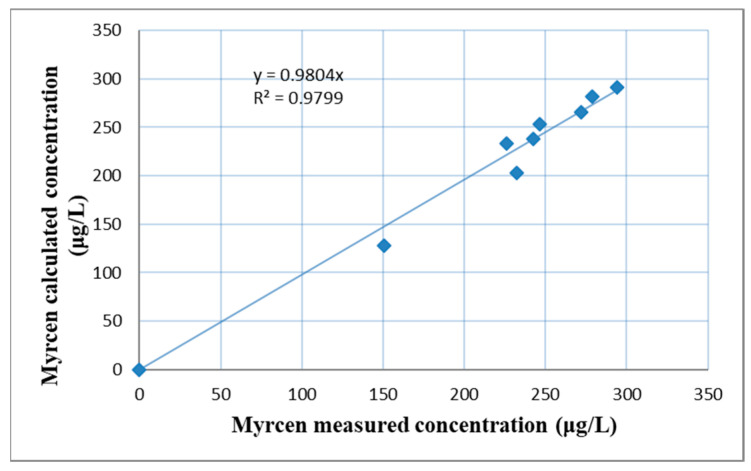
Recalculated and measured β-myrcene concentrations fitted to model.

**Figure 9 plants-11-01043-f009:**
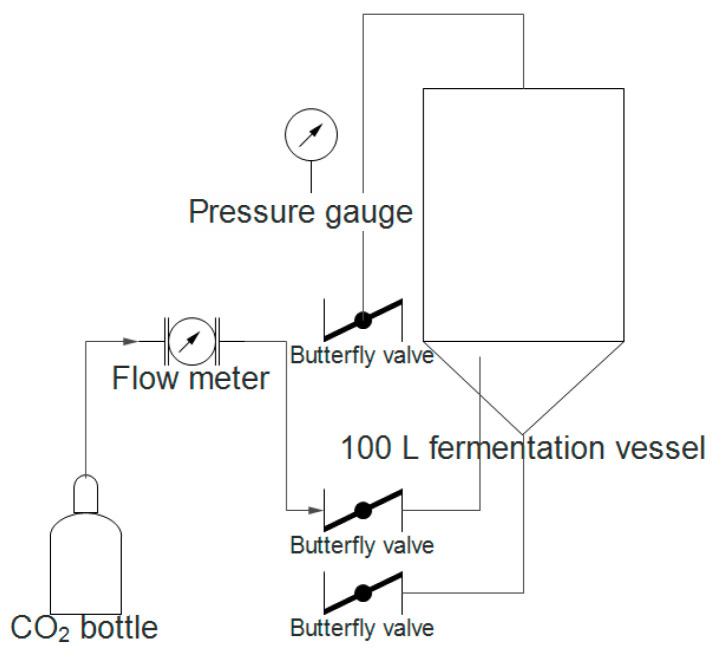
Dry hopping procedure schematic drawing.

**Figure 10 plants-11-01043-f010:**
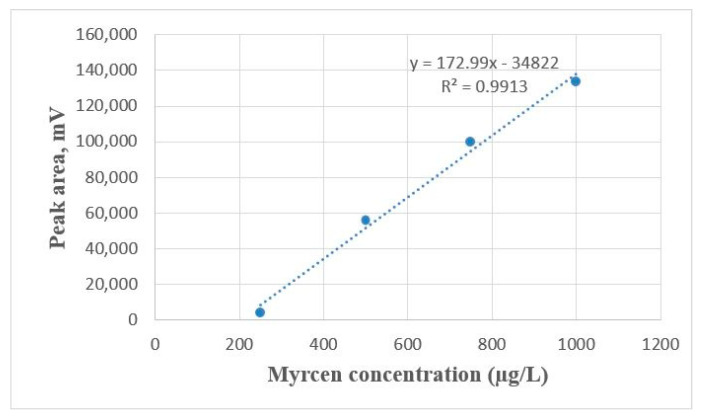
Calibration curve of β-myrcene concentration.

**Figure 11 plants-11-01043-f011:**
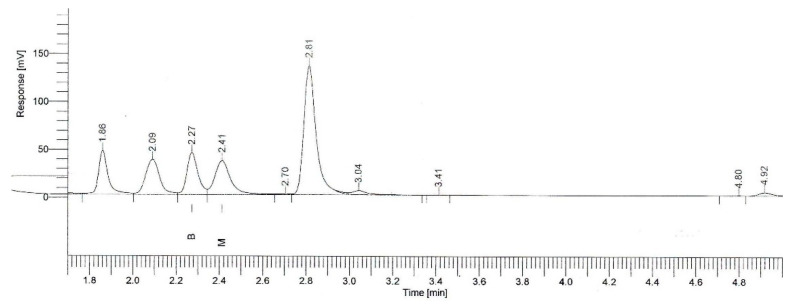
Typical chromatogram for β-myrcene (2.27 min. butanol (B), 2.41 min. β-myrcene (M)).

**Table 1 plants-11-01043-t001:** Main parameters of the beers compared to standards.

Parameters	Young Beer	After Dry Hopping	BJCP Parameters
Original extract (% *w*/*w*)	15.15 ± 0.03 ^a^*	15.27 ± 0.02 ^a^	-
Final extract (% *w*/*w*)	5.79 ± 0.01 ^a^	5.58 ± 0.01 ^a^	-
Original density (g/cm^3^)	1.065 ± 0.001 ^a^	1.067 ± 0.001 ^a^	1.050–1.085
Alcohol (% *v*/*v*)	5.9 ± 0.05 ^a^	5.8 ± 0.05 ^a^	5.5–9.0
Final density (g/cm^3^)	1.025 ± 0.001 ^a^	1.020 ± 0.001 ^a^	1.050–1.085
CO_2_ (g/L)	3.56 ± 0.02 ^a^	4.48 ± 0.03 ^b^	-
O_2_ (mg/L)	0.06 ± 0.01 ^a^	0.22 ± 0.01 ^a^	-
Turbidity (EBC)	11.19 ± 0.03 ^a^	16.32 ± 0.04 ^b^	-
pH	4.59 ± 0.01 ^a^	4.61 ± 0.01 ^a^	-
Color	132.3 ± 0.2 ^a^	131.7 ± 0.1 ^a^	50–80
Bitterness (IBU)	54.5 ± 0.2 ^a^	64.3 ± 0.4 ^b^	50–90
β-myrcene (μg/L)	0 ± 0.00 ^a^	228 ± 0.7 ^b^	

* The results are the mean ± standard deviation (*n* = 3). Beer samples: a–b, mean values in the same row followed by different letters means significant difference (*p* < 0.05).

## Data Availability

The data presented in this study are available in this article.

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
