# Peer review of "The Effect of Dry Hopping Efficiency on β-Myrcene Dissolution into Beer"

_plants, 2022, doi:10.3390/plants11081043_

Round 1

Reviewer 1 Report

The research planned and carried out by the Authors seems to be interesting and promising. Authors have investigated the dissolution rate of the main volatile component of hops, β-myrcene, using modified dry hopping method and determined various important parameters of the beer after the process. They determined, among other things, bitterness, pH, CO2 and alcohol content, extract and density during the conducted process. All sections were correctly described by the Authors. The results were analyzed and discussed. The manuscript can be published in Plants in its present form.

Author Response

2 April 2022

Dear Referee,  

We would like to thank the referee for the close reading of our manuscript and for her/his recommendation to publish the paper entitled The effect of dry hopping efficiency on β-myrcene dissolution into beer in Plants journal.

Referee comments: The research planned and carried out by the Authors seems to be interesting and promising. Authors have investigated the dissolution rate of the main volatile component of hops, β-myrcene, using modified dry hopping method and determined various important parameters of the beer after the process. They determined, among other things, bitterness, pH, CO2 and alcohol content, extract and density during the conducted process. All sections were correctly described by the Authors. The results were analyzed and discussed. The manuscript can be published in Plants in its present form.

Response: Thank you very much for your appreciation and recommendation.

Sincerely,

Georgiana Codină and co.

Reviewer 2 Report

The aim of this study was to determine the amount of β-myrcene volatile compound, the main component of the hop oil during dry hopping. Also the physicochemical changes that occur in beer samples during dry hopping were analyzed and discussed.

Abstract:

Line 28 in literature is replaced as in different studies.

Introduction:

Introduction is very well described, but can be shorter. 

Material and methods:

This part is very well described, but needs more information about the number of tested samples.

Line 368 correct title

Results:

Table 11 center numbers.

Figures 1-4 should be the same size.

Line 140 correct title as . β-myrcene.

Figure 5 center figure.

Discussion needs more information about different studies with comparison of results.

Conclusion:

It is OK. 

Author Response

2 April 2022

Dear Referee,  

We would like to thank the referee for the close reading and for the proper suggestions. We hope that we provide all the answers to the reviewer’s comments.

Thank you very much for the recommendations to publish our paper entitled “The effect of dry hopping efficiency on β-myrcene dissolution into beer”.

The present version of the paper has been revised according to the reviewer’s suggestions.             

Referee comments: The aim of this study was to determine the amount of β-myrcene volatile compound, the main component of the hop oil during dry hopping. Also the physicochemical changes that occur in beer samples during dry hopping were analyzed and discussed.

Response: We would like to thank to the referee for her/his close reading of our manuscript.

Referee comments: Line 28 in literature is replaced as in different studies.

Response: We corrected in the manuscript.

Referee comments: Introduction is very well described, but can be shorter. 

Response: We would like to thank to the referee for her/his appreciations. We did make some modification to the introduction part.

Referee comments: Material and methods: This part is very well described, but needs more information about the number of tested samples.

Response: We completed the materials and methods part according to the referee suggestions. We mentioned now that each sample was analyzed three times and we presented in a more detailed way the method of determining the myrcene.

Referee comments: Line 368 correct title

Response: We corrected the title.

Referee comments: Table 11 center numbers

Response: We centered them.

Referee comments: Figures 1-4 should be the same size

Response: We tried to optimize them.

Referee comments: Line 140 correct title as . β-myrcene

Response: We corrected the title.

Referee comments: Figure 5 center figure.

Response: We tried to optimize it.

Referee comments: Discussion needs more information about different studies with comparison of results.

Response: We completed the discussion part according to referee suggestions.

Referee comments: Conclusion: It is OK. 

Response: Thank you very much for your appreciation.

Sincerely,

Georgiana Codină et al.

Reviewer 3 Report

In my opinion, the article “The effect of dry hopping efficiency on β-myrcene dissolution into beer” needs a major revision. Topic is relatively interesting, results seem relatively novel, although the amount of work is not very high, and the editor can consider the publication (after revision and if accepted) as a short communication. The Abstract and the Introduction were written much better than Materials and methods and results and Discussion, which have to be revised more significantly since these parts are not at the level of Plants journal. English should also be revised in these parts, and sometimes readers cannot conclude if authors are talking about their results or something from the literature.

Specific comments:

L68-71: Please revise this, correct the syntax, the sentence is awkward, I think there was some mistake.

L91-92: Same as above, revise this, correct the syntax, the sentence is awkward, I think there was some mistake (“IPA beer is considered critical oil”).

L98: Explain an abbreviation when first mentioned in the text: NTP.

Lines are wrong in the rest of the manuscript (PDF)…

Page 3

L10: Explain what is bw.

“Quantitative and …. were analyzed and discussed.” – Authors should better state the aim of this study, why it is important, the novelty, contribution to the field, etc. For example, why B-myrcene is so important, it is not very clear from the introduction since it is always below the odor detection threshold – please make it clearer and be more concise.

L10: Explain what is BCJP, not only in the footnotes of the table. Who is the author of this program, who authorized it? Include a reference.

L11: Table 1 footnote: this is not correct, values in the same row are significantly different. Please use always a and b, there is no need to use k, l, etc.

L12: Other researchers, while only a single reference is included (23). Please include more references, in general, in the whole manuscript, about beer aroma and similar.

L12: Values 3.5 and 4.6 are not exactly the same as those reported in the table, please revise or explain.

Figure 3 and similar: Authors stated the change is insignificant, please include error bars in the graphs.

Page 4

Please update the titles of figures and tables - from the titles it is not clear that it is about beer. The titles of tables and figures should be self-informative without consulting the text, please correct them in this way. Please use decimal points instead of commas.

L13: Alcohol concentration increase of 7% - it was written in a way it is not clear if this was a result obtained in this study or something from the literature.

L14: In triplicate – was the experiment set in triplicate or just sampling for B-myrcene determination? What about other analyses? Please clarify this.

Page 5

L15: 3760 microg/L – how was this determined, is it from literature? If so, please provide a reference.

Page 6

Paragraph 2.5: Most of what is written does not make much sense. In my opinion, most important results from Figure 9 could be the difference of b-myrcene and bitterness between the final and initial samples, while all other parameters are grouped one next to the other (initial / final). This would mean that it was confirmed that dry hoping caused a change in these two parameters mostly.

Please use past tense when discussing your own results:

L21: “…after hopping are insignificant ones” (use “were” instead of “are” if these are you results you are referring)

L21: “…This will lead to a decrease…” (use “This led to a decrease…” if these are you results you are referring).

Page 8

First half of Page 8: Please use references when talking about processes and changes in beer which are obviously clips form the literature (physical absorption, large surface, turbidity, proteins, tannins, etc.).

L25: “The pH value does not vary significantly…” use past tense throughout the manuscript!

L26: “…and not within the lupulin gland but…” what is this, it seems a little bit out of context, please explain.

L26: “B-Myrcene were…” – please use singular – revise English throughout the article!

L26-27: Please use the full name of B-myrcene when first mentioned, not here.

L27: “entrained” – I am not sure what did authors mean, maybe “removed”? Please use another word or explain better.

Page 9

L28-30: This part again, like in the results section, seems redundant: In my opinion, most important results from Figure 9 could be the difference of b-myrcene and bitterness between the final and initial samples, while all other parameters are grouped one next to the other (initial / final). This would mean that it was confirmed that dry hoping caused a change in these two parameters mostly.

Paragraph 4.1: “Based on our previous works…” please add citations.

L32-33: Do not start a sentence with a number, use “Ten g…” instead of “10 g…” when starting a sentence – throughout the text.

Page 10

Figure 10: Please increase the size of the figure and especially letters.

L35-37: Please add references for all the standard methods used. For all equipment, please state a model and then in parentheses company, city and country.

Page 11

L37: Are you sure the hydrogen flow was so high? Maybe not the one through the column? Please check and revise if necessary.

L37-39: The preparation of standard solutions is not clearly described. For example, which solution and for what it was used when 18.8 microg/L of myrcene stock solution was added? Is it another stock solution, I think not? It could be the description of the way standard solutions were prepared, but in this case it is not always 18.8 microg/L of myrcene solution pipetted. Please mention stock solution before standard solution. As well, there were 4 x 100 mL solutions, but only three volumes added in the parenthesis.

Were analyses replicated? It seems the calibration curve was prepared, but it seems there was only a single analysis of each standard concentration level. Was everything corrected to the IS? Please add more details. Was there any kind of method validation, recovery, repeatability? Please add these data.

Author Response

2 April 2022

Dear Referee,  

We would like to thank the referee for the close reading and for the proper suggestions. We hope that we provide all the answers to the reviewer’s comments.

Thank you very much for the recommendations to publish our paper entitled “The effect of dry hopping efficiency on β-myrcene dissolution into beer”.

The present version of the paper has been revised according to the reviewer’s suggestions.             

Referee comments: In my opinion, the article “The effect of dry hopping efficiency on β-myrcene dissolution into beer” needs a major revision. Topic is relatively interesting, results seem relatively novel, although the amount of work is not very high, and the editor can consider the publication (after revision and if accepted) as a short communication. The Abstract and the Introduction were written much better than Materials and methods and results and Discussion, which have to be revised more significantly since these parts are not at the level of Plants journal. English should also be revised in these parts, and sometimes readers cannot conclude if authors are talking about their results or something from the literature.

Response: We would like to thank to the referee for her/his close reading of our manuscript and appreciation. The results and subject is a novel one very little discusses in the international literature. The methodology used is a complex one and we want to mention that the beer is a product that takes time to obtain it and that is not too easy to obtain it like for example bakery and dairy products one. We tried to improve the entire manuscript according to the referee suggestions in especially the discussion part as referee suggested. Also we completed the manuscript with more references used to argue our results obtained.

Specific comments:

Referee comments: L68-71: Please revise this, correct the syntax, the sentence is awkward, I think there was some mistake.

Response: We corrected and revised.

Referee comments: L91-92: Same as above, revise this, correct the syntax, the sentence is awkward, I think there was some mistake (“IPA beer is considered critical oil”).

Response: We corrected and revised.

Referee comments: Explain an abbreviation when first mentioned in the text: NTP.

Response: We explained.

Referee comments: Page 3. L10: Explain what is bw.

Response: We explained.

Referee comments: Page 3: “Quantitative and …. were analyzed and discussed.” – Authors should better state the aim of this study, why it is important, the novelty, contribution to the field, etc. For example, why B-myrcene is so important, it is not very clear from the introduction since it is always below the odor detection threshold – please make it clearer and be more concise.

Response: We reformulated according to referee suggestions.

Referee comments: Page 3. L10: Explain what is BCJP, not only in the footnotes of the table. Who is the author of this program, who authorized it? Include a reference.

Response: We explained and we included a reference.

Referee comments: Page 3. L11: Table 1 footnote: this is not correct, values in the same row are significantly different. Please use always a and b, there is no need to use k, l, etc.

Response: We revised according to the referee suggestions.

Referee comments: Page 3. L12: Other researchers, while only a single reference is included (23). Please include more references, in general, in the whole manuscript, about beer aroma and similar.

Response: We added more references not only on this line but also in whole manuscript about beer aroma and similar according to the referee suggestions.

Referee comments: Page 3. L12: Values 3.5 and 4.6 are not exactly the same as those reported in the table, please revise or explain.

Response: We corrected.

Referee comments: Figure 3 and similar: Authors stated the change is insignificant, please include error bars in the graphs.

Response: We included.

Referee comments: Page 4. Please update the titles of figures and tables - from the titles it is not clear that it is about beer. The titles of tables and figures should be self-informative without consulting the text, please correct them in this way. Please use decimal points instead of commas.

Response: We revised and corrected.

Referee comments: Page 4. L13: Alcohol concentration increase of 7% - it was written in a way it is not clear if this was a result obtained in this study or something from the literature.

Response: Was from literature. We added the reference according to referee suggestions.

Referee comments: Page 4. L14: In triplicate – was the experiment set in triplicate or just sampling for B-myrcene determination? What about other analyses? Please clarify this.

Response: We clarified and we added the methods.

Referee comments: Page 5. L15: 3760 microg/L – how was this determined, is it from literature? If so, please provide a reference.

Response: We completed, it is the concentration calculated from the hop analysis bulletins that producer declared.

Referee comments: Page 6. Paragraph 2.5: Most of what is written does not make much sense. In my opinion, most important results from Figure 9 could be the difference of b-myrcene and bitterness between the final and initial samples, while all other parameters are grouped one next to the other (initial / final). This would mean that it was confirmed that dry hoping caused a change in these two parameters mostly.

Response: We agree with the referee point of view. We added the figure due to make a more statistical approach to our results. But as referee suggested has no significant relevance for our study so we agree to delete it from the manuscript.

Referee comments: Please use past tense when discussing your own results: L21: “…after hopping are insignificant ones” (use “were” instead of “are” if these are you results you are referring); L21: “…This will lead to a decrease…” (use “This led to a decrease…” if these are you results you are referring).

Response: We revised.

Referee comments: First half of Page 8: Please use references when talking about processes and changes in beer which are obviously clips form the literature (physical absorption, large surface, turbidity, proteins, tannins, etc.).

Response: We added references and also we completed the discussion part according to the referee suggestions.

Referee comments: Page 8: L25: “The pH value does not vary significantly…” use past tense throughout the manuscript!

Response: We corrected.

Referee comments: Page 8: L26: “…and not within the lupulin gland but…” what is this, it seems a little bit out of context, please explain.

Response: We agree with the referee point of view. Was a little out of context. We deleted this part.

Referee comments: Page 8: L26: “B-Myrcene were…” – please use singular – revise English throughout the article!

Response: We corrected.

Referee comments: Page 8: L26-27: Please use the full name of B-myrcene when first mentioned, not here.

Response: We corrected.

Referee comments: Page 8: L27: “entrained” – I am not sure what did authors mean, maybe “removed”? Please use another word or explain better.

Response: We corrected.

Referee comments: Page 9: L28-30: This part again, like in the results section, seems redundant: In my opinion, most important results from Figure 9 could be the difference of b-myrcene and bitterness between the final and initial samples, while all other parameters are grouped one next to the other (initial / final). This would mean that it was confirmed that dry hoping caused a change in these two parameters mostly.

Response: We agree with the referee point of view. The discussion related to the figure 9 has no significant relevance for our study so we agree to delete the figure and the discussion part about it from the manuscript.

Referee comments: Page 9: Paragraph 4.1: “Based on our previous works…” please add citations.

Response: We do not have any published only laboratory trials. That way we wrote in this form. We revised now.

Referee comments: Page 9: L32-33: Do not start a sentence with a number, use “Ten g…” instead of “10 g…” when starting a sentence – throughout the text.

Response: We corrected.

Referee comments: Page 10, Figure 10: Please increase the size of the figure and especially letters.

Response: We revised.

Referee comments: Page 10. L35-37: Please add references for all the standard methods used. For all equipment, please state a model and then in parentheses company, city and country.

Response: We introduced.

Referee comments: Page 11. L37: Are you sure the hydrogen flow was so high? Maybe not the one through the column? Please check and revise if necessary.

Response: Yes, is correct.

Referee comments: Page 11.  L37-39: The preparation of standard solutions is not clearly described. For example, which solution and for what it was used when 18.8 microg/L of myrcene stock solution was added? Is it another stock solution, I think not? It could be the description of the way standard solutions were prepared, but in this case it is not always 18.8 microg/L of myrcene solution pipetted. Please mention stock solution before standard solution. As well, there were 4 x 100 mL solutions, but only three volumes added in the parenthesis; Were analyses replicated? It seems the calibration curve was prepared, but it seems there was only a single analysis of each standard concentration level. Was everything corrected to the IS? Please add more details. Was there any kind of method validation, recovery, repeatability? Please add these data.

Response: We want to thank to the referee for his/her close reading of our manuscript. We corrected and reformulated in order to be easier to understand. We did several measurements for the standards and in several days, we corrected the results with IS. We offered more details in the manuscript now according to the referee suggestions. 

Sincerely,

Georgiana Codină et al.

Round 2

Reviewer 2 Report

Authors accepted all comments.

Author Response

9 April 2022

Dear Referee,

We would like to thank the referee for the close reading and for the proper suggestions. We hope that we provide all the answers to the reviewer’s comments.

Thank you very much for the recommendations to publish our paper entitled The effect of dry hopping efficiency on β-myrcene dissolution into beer.

The present version of the paper has been revised according to the reviewer’s suggestions.             

We uploaded the corrected version of the article for which we used the red color (for English language corrections) for the addition text.

REFERE COMMENTS:

Reviewer Comments

Authors accepted all comments.

Response: We would like to thank to the referee for his/her close reading of our manuscript and for his/her appreciation. Also we want to thank to his/her recommendation to publish our manuscript in Plants journal.

Sincerely,

Georgiana Codina et co.

Reviewer 3 Report

In my opinion the manuscript is now significantly improved and the majority of the recommendations have been adopted by the authors. With a short explanation in the introduction, authors significantly increased the apparent importance of the study (health aspects of B-myrcene). The manuscript still needs some minor corrections, as follows.

Although the text is understandable, English could be improved, I suggest a revision by a native speaker. New parts are written well, but some parts from the original version require some improvements.

When you starting a sentence with β-myrcene, please put capital letters M, such as β-Myrcene…

Figure 4: Explain the abbreviations Alex and DMA in the figure title

L183: Please delete the year 2013

L225-226: “This will lead to a decrease of the final extract up to 3.76% during dry hopping…“ – I still cannot conclude if this is a result from literature or your result, please make it clearer.

L235: “Moreover, the fermentation temperature decreases from 18°C to 8°C,…” – literature or your result?

L287: Please add full chemical name of B-myrcene (7-methyl-3-methylideneocta-1,6-diene) when first mentioned in the introduction and not here, like I suggested in my previous review.

L303: Please correct: “…on the method, [6, 7] used volume…”

L341: “Ten gramS…”, plural

L343: “Seven gramS…”, plural

Please revise slightly the Conclusions, English is a bit awkward.

Author Response

9 April 2022

Dear Referee,

We would like to thank the referee for the close reading and for the proper suggestions. We hope that we provide all the answers to the reviewer’s comments.

Thank you very much for the recommendations to publish our paper entitled The effect of dry hopping efficiency on β-myrcene dissolution into beer.

The present version of the paper has been revised according to the reviewer’s suggestions.             

We uploaded the corrected version of the article for which we used the red color for the addition text.

REFERE COMMENTS:

Reviewer Comments

In my opinion the manuscript is now significantly improved and the majority of the recommendations have been adopted by the authors. With a short explanation in the introduction, authors significantly increased the apparent importance of the study (health aspects of B-myrcene). The manuscript still needs some minor corrections, as follows.

Response: We would like to thank to the referee for his/her close reading of our manuscript and for his/her appreciation. Also we want to thank to his/her recommendation to publish our manuscript in Plants journal.

Reviewer: Although the text is understandable, English could be improved, I suggest a revision by a native speaker. New parts are written well, but some parts from the original version require some improvements.

Response: We would like to thank to the referee for his/her close reading of our manuscript. All the article was corrected from the English point of view by an English teacher.

Reviewer: When you starting a sentence with β-myrcene, please put capital letters M, such as β-Myrcene…

Response: We corrected in the manuscript according to the referee suggestions.

Reviewer: Figure 4: Explain the abbreviations Alex and DMA in the figure title

Response: We explained in the manuscript according to the referee suggestions.

Reviewer: L183: Please delete the year 2013

Response: We deleted.

Reviewer: L225-226: “This will lead to a decrease of the final extract up to 3.76% during dry hopping…“ – I still cannot conclude if this is a result from literature or your result, please make it clearer.

Response: Was our data. We revised in the manuscript according to the referee suggestions.

Reviewer: L235: “Moreover, the fermentation temperature decreases from 18°C to 8°C,…” – literature or your result?

Response: Was our data. We revised in the manuscript according to the referee suggestions.

Reviewer: L287: Please add full chemical name of B-myrcene (7-methyl-3-methylideneocta-1,6-diene) when first mentioned in the introduction and not here, like I suggested in my previous review.

Response: We added.

Reviewer: L303: Please correct: “…on the method, [6, 7] used volume…”

Response: We corrected.

Reviewer: L341: “Ten gramS…”, plural

Response: We corrected.

Reviewer: L343: “Seven gramS…”, plural

Response: We corrected.

Reviewer: Please revise slightly the Conclusions, English is a bit awkward.

Response: We revised the conclusion part and we want to thank to the reviewer for his/her suggestions.

Sincerely,

Georgiana Codina et co.
